# Coastal Scenic Quality Assessment of Moroccan Mediterranean Beaches: A Tool for Proper Management

**Noureddine Er-Ramy** [1] , **Driss Nachite** [1] , **Giorgio Anfuso** [2,*] and **Allan T. Williams** [3]

1    Department of Geology, Faculty of Sciences, University of Abdelmalek Essaadi, Tetouan 93000, Morocco;
     ramy.nourdin@gmail.com (N.E.-R.); dnachite@uae.ac.ma (D.N.)
2    Department of Earth Sciences, Faculty of Marine Sciences, University of Cádiz, 11510 Puerto Real, Spain
3    Faculty of Architecture, Computing and Engineering, University of Wales Trinity Saint David (Swansea),
     Mount Pleasant, Swansea SA1 6ED, UK; allanwilliams512@outlook.com
*    Correspondence: giorgio.anfuso@uca.es; Tel.: +34-956-016167

**Abstract:** This paper focuses on the study of landscape quality of Moroccan Mediterranean coastal areas, with a view to distinguishing exceptional beaches with high scenic value. The main characteristics of 50 beaches along the studied coast were assessed using a coastal scenic evaluation system based on a set of 26 selected parameters, including physical (18) and human (8) parameters. Each parameter was examined via a five-point rating scale, ranging from presence/absence or poor quality (1) to excellent quality (5). A decision index (D) is afterward obtained and used to classify sites into five classes: Class I: D ≥ 0.85, which included 9 sites (18%); Class II: 0.85 > D ≥ 0.65, 10 sites (20%); Class III: 0.65 > D ≥ 0.40, 8 sites (16%); Class IV: 0.40 > D ≥ 0.00, 16 sites (32%); and Class V: D < 0.00, 7 sites (14%). The sites of Belyounech 2, Maresdar, El Hwad, and Dalya are the best examples of Class I and represent extremely attractive coastal landscapes. The sites of Ghandouri, Tangier Municipal, M'Diq, Martil, and Tangier Malabata are examples of degraded urban sites that are very unattractive due to high human pressures. Management efforts in Moroccan coastal landscapes can strengthen the control of human activities and improve the scenic value of the sites. Class II beaches, such as Mrisat, Souani, Taourirt, and Sfiha, could improve and upgrade to Class I through litter cleaning and a regular maintenance program. Using the same principle, Class III sites, such as Sidi Amer O Moussa and Sidi Driss, could improve and upgrade to Class II. Indeed, litter and sewage appear as the main factors of degradation of Moroccan coasts, and many excellent beaches are strongly affected by them. This should be a wakeup call to the Moroccan authorities to take urgent and appropriate management measures.

**Keywords:** landscape; tourism; physical and human parameters; litter; Morocco

## 1. Introduction

In a global context of natural site management, conservation, and protection, the study and evaluation of Mediterranean coastal areas is of primary interest since they have high ecological and economic importance. Coastal areas are very dynamic, highly fragile, and sensitive environments, threatened by various human and natural factors [1]. They present a great uniqueness and relevant biological, geological, ecological, and landscape diversity and also constitute a favourable environment for various activities, mainly linked to recreational tourism and other economic activities [2]. Unfortunately, coastal environments are usually highly exploited, being affected by the establishment of infrastructures and human impacts that are often accentuated because of the mismanagement of coastal areas. Presently, the Mediterranean is considered the world's leading tourist destination. According to estimates of the World Tourism Organization [3], Mediterranean destinations recorded 58 million international arrivals in 1978 and will record 500 million in 2030.

Before the coronavirus pandemic (COVID-19), international tourism posted record results; in 2019, the total number of tourists grew, with respect to 2018, by 4% globally,

reaching 1.45 billion visitors and contributing USD 1478 billion in revenue or 10.4% of the global Ground Domestic Product (GDP) [4]. On the Mediterranean scale, the number of tourists increased by 5.8% in 2019 compared to 2018 [4], reinforcing the Mediterranean position around the world as a leading tourist destination. According to the World Tourism and Travel Council [5], in 2019, tourism provided 11.3% of the employment in the region and constituted 11.5% of GDP. In Morocco, tourism activity continues to attract more and more tourists each year: USD 8.4 billion revenue was recorded in 2019 and tourism drove the economic growth with a contribution of 7% to GDP. The number of international tourist arrivals in 2019 reached 12.93 million, with an annual increase of 5%. Further, tourism activity generated a very large number of jobs in 2019, ca. 550,000, corresponding to 5% of the active population [6].

Coastal areas have probably been the most affected areas by tourism developments because they are the most appreciated and favoured places by visitors [7–9]. Coastal tourism is among the most increasing economic activities in all Mediterranean countries and it is considered among the largest active industries in the world [10,11]. It is mostly linked to "Sun, Sea and Sand" (3S) tourism [12,13], and, therefore, beaches present a main player in this industry [7,14]. This is the reason why countries with attractive beaches are the most visited by tourists [9,14,15]. The beach is the main support of seaside tourism, and beautiful attractive beaches around the world are worth billions of US dollars as they contribute extraordinary revenues [16]. Coastal landscapes are very important for beach lovers [17], especially for users interested in natural beaches, and numerous studies have confirmed that one of the most important aspects that tourists take into account when choosing beaches is the excellent coastal scenery [18–20]. In this context, according to Micallef [21], Williams and Micallef [22] and Williams [23], five parameters are extremely important for coastal tourists, which constitute the "Big Five", namely, scenery, water quality, facilities, litter, and safety—their order of importance changing depending on beach typology. Scenery is the objective of this paper. Barbosa de Araujo and Costa [24], working in Brazil, showed that '*landscape was probably highly rated as an attribute in visitor's choice*', and '*the maintenance of the landscape quality of beaches must be the main priority*'. Scenery can be defined as '*the appearance of an area*' ([25], p. 4), '*is part of a coastal landscape inventory available for different coastal zone disciplines*' ([26], p. 18). Another definition ([27], p. 309), considered that '*scenery is a combination of the physical and cultural environment and it can be argued that the former is fixed by nature and only the cultural can be modified by man*'. Similarly, coastal landscapes can be described as '*a littoral area, as perceived by people, whose character results from the numerous interactions of natural and/or human factors*' ([25], p. 32). Scenery has been defined as a central parameter, for decades, in the choice and selection of beaches [28,29]. Coastal scenery is a multifunctional system of high environmental and economic value. Beautiful scenery is a very important element for the sustainable development of coastal tourism and is an engine of economic growth in many Mediterranean coastal countries [30].

Coastal scenery assessment is a fundamental tool for managers to enhance coastal conservation, protection, development and the execution of management plans, and its perception is very important [31]. Results of scenic assessments provide extremely useful objective information and recommendations, allowing the creation of a scientifically sound database [32–35]. In this context, on 12 March 2013, the European Commission [36] launched a new joint initiative on integrated coastal management (ICZM) and maritime spatial planning (MSP), aiming to establish a framework for maritime spatial planning and integrated coastal management in EU Member States to promote sustainable growth of maritime and coastal activities and the sustainable use of coastal and marine resources. Coherent application of MSP and ICZM will improve the interaction between land- and sea-based activities. Coastal scenery characterization and protection fit well into this strategy as they are focused on sustainably managing coastal resources in an integrated and informed way, while protecting the environment that has been experiencing numerous growing pressures in the last decades. It is essentially a management process that facilitates

a more integrated working partnership of the different key interests, especially including local communities.

Since coastal scenery is a valuable resource [30], assessment has to be carried out in a scientific, objective, and practical way, and this represents a challenge. The main objectives of this paper are (i) the characterization of 50 beaches along the Moroccan Mediterranean coast, according to their typology (urban, resort, etc. [22,23]); (ii) the assessment of their scenic value through selected variables based on a scientific methodology, i.e., the Coastal Scenic Evaluation System (CSES, [37]); and (iii) to provide indications to preserve and enhance coastal scenic beauty.

At the same time, this paper aims to follow up the coastal scenic quality of several sites that were evaluated 10 years previously by Khattabi et al. [38] and by Williams and Khattabi [39], with the same methodology adopted in this study. In addition, it analyses the different sources of impacts and presents complementary information necessary and useful for any future sound management plan.

## 2. Study Area

Morocco has a special geographical position, located at the north-western end of the African continent, between the Atlantic Ocean and the Mediterranean Sea (Figure 1). The Moroccan coastline is characterized by two large maritime façades extending up to 3500 km, with a maritime area >1 million km$^2$. The Atlantic Ocean coast extends over approximately 3000 km and the Mediterranean one is ca. 512 km in length. Morocco has 9 coastal regions, with 145 beaches that have been declared official bathing areas [40]. Along the Mediterranean coast, cliffs occupy nearly 80% of the coast length, while beaches represent 20% [41].

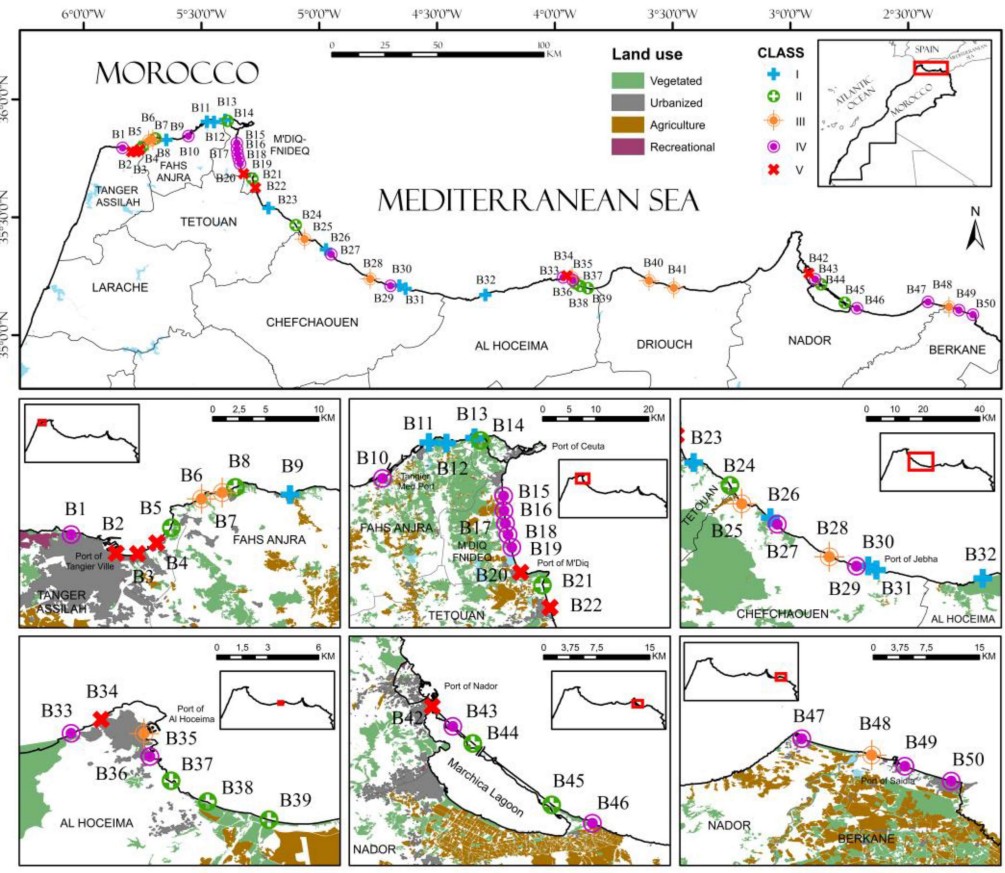

**Figure 1.** Location map of the sites studied with indications of their land use and scenic class.

The Moroccan Mediterranean coastline studied in this paper is home to a rich and varied natural heritage, characterized by a rich landscape and an important morphological diversity, such as sandy beaches, cliffs, coves, lagoons, deltas, marshes, and fixed and mobile dunes. The 50 Moroccan Mediterranean beaches selected (Figure 1) are located between Cap Spartel (Tangier) at the westernmost end, and Saïdia at the easternmost end. The sites selected constitute the most frequented beaches and are among the most representative of the characteristics of the Moroccan Mediterranean coastline.

## 3. Methods

Main beaches characteristics along the Moroccan Mediterranean coast were assessed using the 'Coastal Scenic Evaluation System' (CSES). This method was first presented by Ergin et al. [37], and updated lately by Ergin [42], and has been later applied in several case studies in more than 30 countries around the world (e.g., UK, USA, Spain, Turkey, Cuba, Ireland, Colombia, Chile, and Brasil). There are five principal steps in the CSES methodology (Figure 2), which will be presented in the following lines; for further information, consult Ergin et al. [37].

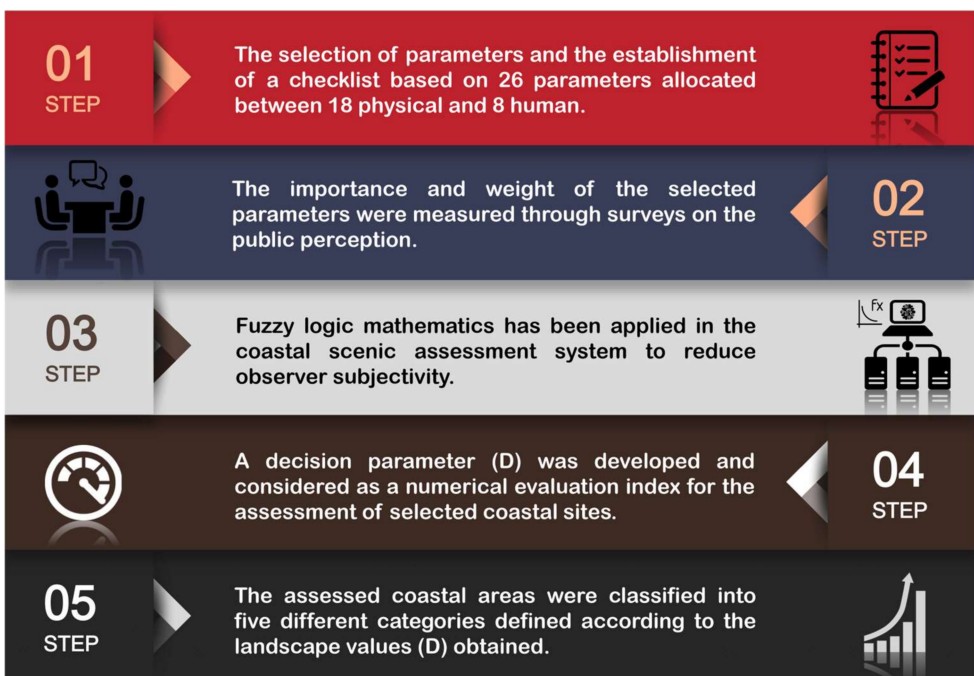

**Figure 2.** Schematic representation of the main steps of the methodology used (CSES).

### Step 1—Choice and selection of scenic parameters

The coastal scenic evaluation system was developed to form a weighted parameter checklist required for coastal landscape assessment. Determination of a checklist of different parameters was the initial step (Table 1). The selection of parameters was carried out through a series of evaluation processes, including a 3-year intellectual study of literature reviews, public enquires to >1000 bathing areas users in various Mediterranean countries, asking what was important for coastal scenic quality [43], and consultation with landscape experts. Finally, twenty-six parameters were identified and classified into two groups of parameters: 18 natural and 8 human [37]. The checklist was successfully field tested in many countries [44].

**Table 1.** Checklist parameters according to the CSES methodology [37,42].

| N° | Physical Parameters | | Rating | | | | |
|---|---|---|---|---|---|---|---|
| | | | 1 | 2 | 3 | 4 | 5 |
| 1 | Cliff | Height (H) | Absent (<5 m) | 5 ≤ H < 30 m | 30 ≤ H < 60 m | 60 ≤ H < 90 m | H ≥ 90 m |
| 2 | | Slope | <45° | 45–60° | 60–75° | 75–85° | Circa vertical |
| 3 | | Features * | Absent | 1 | 2 | 3 | Many > 3 |
| 4 | Beach face | Type | Absent | Mud | Cobble/Boulder | Pebble/Gravel | Sand |
| 5 | | Width (W) | Absent | W < 5 or W > 100m | 5 ≤ W < 25 m | 25 ≤ W < 50 m | 50 ≤ W ≤ 100 m |
| 6 | | Color | Absent | Dark | Dark tan | Light tan/bleached | White/gold |
| 7 | Rocky shore | Slope | Absent | <5° | 5–10° | 10–20° | >20° |
| 8 | | Extent | Absent | <5 m | 5–10 m | 10–20 m | >20 m |
| 9 | | Roughness | Absent | Distinctly jagged | Deeply pitted and/or irregular | Shallow pitted | Smooth |
| 10 | Dunes | | Absent | Remnants | Fore-dune | Secondary ridge | Several |
| 11 | Valley | | Absent | Dry | Stream (<1 m) | Stream (1–4 m) | >4 m |
| 12 | Skyline landforms | | Not visible | Flat | Undulating | Highly undulating | Mountainous |
| 13 | Tides | | Macro (>4 m) | | Meso (2–4 m) | | Micro (<2 m) |
| 14 | Coastal landscape features ** | | None | 1 | 2 | 3 | >3 |
| 15 | Vistas | | Open on one side | Open on two sides | | Open on three sides | Open on four sides |
| 16 | Water colour & clarity | | Muddy Brown/grey | Milky blue/green; opaque | Green/grey blue | Clear blue/dark blue | Very clear turquoise |
| 17 | Vegetation cover | | Bare (<10% vegetation only) | Scrub/Garigue/grass (marram/ferns, etc) | Wetland/meadow | Coppices, maquis (±mature trees) | Variety of mature trees/natural cover |
| 18 | Vegetation debris | | Continuous >50 cm high | Full strand line | Single accumulation | Few scattered items | None |
| | **Human parameters** | | | | | | |
| 19 | Disturbance factor | | Intolerable | Tolerable | | Little | None |
| 20 | Litter | | Continuous accumulations | Full strand line | Single accumulation | Few scattered items | Virtually absent |
| 21 | Sewage (discharge evidence) | | Sewage Evidence | | Some evidence (1–3 items) | | No evidence of sewage |
| 22 | Non-built environment | | None | | Hedgerow/terracing/monoculture | | Mixed cultivation ± trees/natural |
| 23 | Built environment | | Heavy industry | Heavy tourism and/or urban | Light tourism and/or urban and/or sensitive industry | Sensitive tourism and/or urban | Historic and/or none |
| 24 | Access type | | No buffer zone/heavy traffic | Buffer zone/light traffic | | Parking lot visible from coastal area | Parking lot not visible from coastal area |
| 25 | Skyline | | Very unattractive | Unattractive | Sensitively designed | Very sensitively designed | Natural/historic features |
| 26 | Utilities *** | | >3 | 3 | 2 | 1 | None |

* Cliff special features: Indentation, banding, folding, scree, irregular profile. ** Coastal landscape features: Peninsulas, rock ridges, irregular headlands, arches, windows, caves, waterfalls, deltas, lagoons, islands, stacks, estuaries, reefs, fauna, embayment, tombola, etc. *** Utilities: Power lines, pipelines, street lamps, groins, seawalls, revetments, etc.

**Step 2—Public surveys for the determination of scenic parameters relevance**

After the selection of parameters, further surveys to coastal users were carried out [37]. The weight of each one of the 26 parameters was determined through several series of surveys (n ≥ 500); i.e., they do not all have the same relevance, as some are more important than others. Each parameter was evaluated by the respondents, who were asked to rate it on a five-point scale according to its importance/relevance, ranging from 'unimportant' (1 point) to 'extremely important' (5 points) [37].

**Step 3—Fuzzy Logic Assessment**

A mathematical model was adopted using Fuzzy Logic Assessment (FLA) mathematics [45] as an appropriate methodology in order to quantify uncertainties and subjective pronouncements inherited in the assessment parameters; for example, vagueness, uncertainty, and errors. Fuzzy logic is a mathematical analysis tool used for processing data that contains uncertainty and whose purpose is to help eliminate individual subjectivity. Ultimately, after evaluating the parameter scores and entering data into the computer, weighted averages of the attributes, membership degrees, and histograms [37,42] were obtained (see later on in the text), which gives an instantaneous image of the investigated scenery. Practically, the more Attribute 5 is observed, the better the beach scenery is; this corresponds to a right-skewed membership degree curve. Thus, a look at the weighted average immediately indicates the potential ranking of the scenic evaluation.

For more information on the calculation methods and the mathematical formulas, see the references cited in the methodology [32,46–48].

**Step 4—Evaluation Index D**

In this paper, to evaluate the scenic characteristics of the 50 selected sites along the Moroccan Mediterranean coast, each parameter was examined by the first two authors that also recollected information on other site aspects. All parameters, in the field, were attributed a value on a five-point scale from one to five, ranging from the presence/absence or poor quality (1) of the element concerned, to excellent quality (5) [37].

As an example, the beach-face type parameter has five attributes: absent, mud, cobble/boulder, pebble/gravel and sand. A coastal site with cobble beach-face type would be scored a 3; a sand beach-face type site would be scored a 5.

Once the scores were attributed to each parameter in an Excel sheet, a Scenic Evaluation Value (D) was obtained, a decision index calculated from the 'membership degrees' in relation to the attributes, which could classify scenic assessment into one of five distinct classes [37,42], varying from Class I (extremely attractive natural sites) to Class V (very unattractive urban sites). The higher the evaluation index D, the higher the quality of the coastal landscape [42].

**Step 5—Coastal scenery classification of sites investigated**

The D evaluation index made it possible to classify each investigated site into one of the five following different classes according to the CSES methodology. Curve break points, based on the midpoint change of slope, allowed a division of sites into the five main classes, where Class I and Class V were within the lowest 15 percentile and top 85 percentile, respectively, when the D criteria percentile values were graphed on a normal plot. Break-point statistical distributions were tested for a Gaussian (normal) distribution [37]. Details of the web access to the computer programme are in Rangel-Buitrago [48].

- Class I: Extremely attractive natural site with a very high landscape value (D ≥ 0.85);
- Class II: Attractive natural site with high landscape value (0.85 > D ≥ 0.65);
- Class III: Mainly natural site with little outstanding landscape features (0.65 > D ≥ 0.4);
- Class IV: Mainly unattractive urban site with a low landscape value (0.4 > D ≥ 0.00);
- Class V: Very unattractive urban site with intensive development and a low landscape value (D < 0).

Last, the investigated sites were divided into different typologies according to the classification of Williams and Micallef [22] and Williams [23]. This method classifies beaches on the degree of anthropogenic influence as remote, rural, village, urban, and resort, considering criteria such as accessibility, environmental conditions, habitation/accommodation level, and community services, with the purpose of demonstrating the method and ease of application to diverse coastal environments. For more details about this type of classification, see the papers mentioned above [22,23].

## 4. Results and Discussion

The beaches investigated were classified according to the evaluation index D (Table 2, Figures 3 and 4). The final evaluation matrices are presented graphically as histograms

(Figure 5A), membership degree of attributes (Figure 5B), and weighted average of attributes (Figure 5C) and according to beach typology (Figure 5D). Histograms give a clear idea of the state of natural and human parameters of a site, and allow an instant assessment of attributes between high or low values, while the weight averages of the attributes delineated the relative comparison of the natural and human parameters [34,37,49].

**Table 2.** Location and main characteristics of the investigated sites along the Moroccan Mediterranean coast.

| Rank | Code | Site | Type | Location | Geographical Coordinates | | D | Class |
|---|---|---|---|---|---|---|---|---|
| 1 | B13 | Belyounech 2 | VILLAGE | M'DIQ-FNIDEQ | 35°54′39.80″ N | 5°23′49.38″ O | 1.12 | I |
| 2 | B32 | Bades | RURAL | AL HOCEIMA | 35°10′13.77″ N | 4°17′41.46″ O | 1.06 | I |
| 3 | B9 | Oued Aliane | RURAL | FAHS-ANJRA | 35°49′39.40″ N | 5°38′58.98″ O | 1.00 | I |
| 4 | B30 | Jebha (Maresdar) | REMOTE | CHEFCHAOUEN | 35°12′35.45″ N | 4°39′29.00″ O | 0.98 | I |
| 5 | B11 | Dalya | RURAL | FAHS-ANJRA | 35°54′18.89″ N | 5°28′37.33″ O | 0.96 | I |
| 6 | B26 | Stehat 2 | REMOTE | CHEFCHAOUEN | 35°21′22.41″ N | 4°57′51.82″ O | 0.93 | I |
| 7 | B12 | Oued El Marsa | RURAL | FAHS-ANJRA | 35°54′15.49″ N | 5°26′47.40″ O | 0.92 | I |
| 8 | B23 | Amsa | RURAL | TETOUAN | 35°32′30.96″ N | 5°13′8.05″ O | 0.90 | I |
| 9 | B31 | Jebha (El Hwad) | REMOTE | CHEFCHAOUEN | 35°12′34.63″ N | 4°38′52.06″ O | 0.88 | I |
| 10 | B38 | Sfiha | RURAL | AL HOCEIMA | 35°12′37.66″ N | 3°54′0.66″ O | 0.82 | II |
| 11 | B5 | Mrisat | RURAL | TANGER-ASILAH | 35°47′59.54″ N | 5°45′2.34″ O | 0.81 | II |
| 12 | B39 | Souani | RURAL | AL HOCEIMA | 35°11′58.36″ N | 3°51′47.80″ O | 0.79 | II |
| 13 | B14 | Belyounech 1 | VILLAGE | M'DIQ-FNIDEQ | 35°54′33.55″ N | 5°23′37.30″ O | 0.78 | II |
| 14 | B8 | Sidi Kankouche 2 | RURAL | FAHS-ANJRA | 35°49′53.71″ N | 5°42′4.70″ O | 0.77 | II |
| 15 | B21 | Cabo Negro | RESORT | M'DIQ-FNIDEQ | 35°39′31.04″ N | 5°16′57.26″ O | 0.76 | II |
| 16 | B37 | Isli | RURAL | AL HOCEIMA | 35°13′11.52″ N | 3°54′47.10″ O | 0.71 | II |
| 17 | B24 | Oued Laou | VILLAGE | TETOUAN | 35°27′45.00″ N | 5° 5′52.35″ O | 0.71 | II |
| 18 | B45 | Taourirt | RURAL | NADOR | 35°7′26.54″ N | 2°44′44.88″ O | 0.69 | II |
| 19 | B44 | Boqueronesa East | RURAL | NADOR | 35°13′17.33″ N | 2°52′37.68″ O | 0.68 | II |
| 20 | B41 | Sidi Amer O Moussa | RURAL | DRIOUCH | 35°12′12.40″ N | 3°30′12.53″ O | 0.64 | III |
| 21 | B40 | Sidi Driss | RURAL | DRIOUCH | 35°13′45.07″ N | 3°35′42.15″ O | 0.60 | III |
| 22 | B7 | Sidi Kankouche 1 | RURAL | FAHS-ANJRA | 35°49′47.52″ N | 5°42′24.40″ O | 0.55 | III |
| 23 | B48 | Saïdia Med West | RESORT | BERKANE | 35°6′58.88″ N | 2°18′58.65″ O | 0.50 | III |
| 24 | B28 | Amtar | RURAL | CHEFCHAOUEN | 35°14′38.63″ N | 4°47′21.26″ O | 0.46 | III |
| 25 | B25 | Kaa Asrass | RURAL | CHEFCHAOUEN | 35°24′35.93″ N | 5°3′53.35″ O | 0.45 | III |
| 26 | B6 | Playa Blanca | RURAL | FAHS-ANJRA | 35°49′28.64″ N | 5°43′22.40″ O | 0.43 | III |
| 27 | B35 | Quemado | URBAN | AL HOCEIMA | 35°14′38.50″ N | 3°55′35.08″ O | 0.42 | III |
| 28 | B1 | Marqala | URBAN | TANGER-ASILAH | 35°47′39.10″ N | 5°50′4.30″ O | 0.39 | IV |
| 29 | B15 | Rifienne | RESORT | M'DIQ-FNIDEQ | 35°48′34.56″ N | 5°21′4.25″ O | 0.36 | IV |
| 30 | B27 | Stehat 1 | VILLAGE | CHEFCHAOUEN | 35°20′40.94″ N | 4°57′6.96″ O | 0.34 | IV |
| 31 | B16 | Almina | RESORT | M'DIQ-FNIDEQ | 35°47′24.26″ N | 5°20′58.15″ O | 0.32 | IV |
| 32 | B49 | Saïdia Med East | RESORT | BERKANE | 35°6′13.85″ N | 2°16′49.11″ O | 0.31 | IV |
| 33 | B19 | Kabila | RESORT | M'DIQ-FNIDEQ | 35°43′36.21″ N | 5°20′16.76″ O | 0.31 | IV |
| 34 | B17 | Restinga | RESORT | M'DIQ-FNIDEQ | 35°45′50.46″ N | 5°20′43.20″ O | 0.31 | IV |
| 35 | B43 | Boqueronesa West | RURAL | NADOR | 35°14′55.67″ N | 2°54′28.26″ O | 0.26 | IV |
| 36 | B33 | Izdhi | URBAN | AL HOCEIMA | 35°14′42.20″ N | 3°57′49.48″ O | 0.23 | IV |
| 37 | B36 | Cala Bonita | URBAN | AL HOCEIMA | 35°14′4.61″ N | 3°55′22.46″ O | 0.21 | IV |
| 38 | B46 | Kariat Arekmane | RESORT | NADOR | 35°6′53.63″ N | 2°43′10.56″ O | 0.20 | IV |
| 39 | B47 | Ras El Ma | VILLAGE | NADOR | 35°8′24.41″ N | 2°24′49.75″ O | 0.16 | IV |
| 40 | B10 | Ksar Sghir | VILLAGE | FAHS-ANJRA | 35°50′38.83″ N | 5°33′21.78″ O | 0.14 | IV |
| 41 | B18 | Marina Smir | RESORT | M'DIQ-FNIDEQ | 35°44′47.60″ N | 5°20′31.51″ O | 0.13 | IV |
| 42 | B29 | Jebha (Zamana) | VILLAGE | CHEFCHAOUEN | 35°12′15.85″ N | 4°40′47.96″ O | 0.12 | IV |
| 43 | B50 | Saïdia | URBAN | BERKANE | 35°5′16.47″ N | 2°13′42.56″ O | 0.09 | IV |
| 44 | B3 | Tangier Malabata | URBAN | TANGER-ASILAH | 35°46′41.00″ N | 5°46′41.41″ O | −0.21 | V |
| 45 | B22 | Martil | URBAN | M'DIQ-FNIDEQ | 35°37′28.24″ N | 5°16′20.12″ O | −0.27 | V |
| 46 | B20 | M'Diq | URBAN | M'DIQ-FNIDEQ | 35°41′6.90″ N | 5°19′17.42″ O | −0.29 | V |
| 47 | B34 | Sabadia | URBAN | AL HOCEIMA | 35°14′54.78″ N | 3°57′22.64″ O | −0.31 | V |
| 48 | B2 | Tangier Municipal | URBAN | TANGER-ASILAH | 35°46′40.28″ N | 5°47′45.01″ O | −0.32 | V |
| 49 | B4 | Ghandouri | URBAN | TANGER-ASILAH | 35°47′15.04″ N | 5°45′42.99″ O | −0.35 | V |
| 50 | B42 | Miami | URBAN | NADOR | 35°15′41.94″ N | 2°55′13.20″ O | −0.50 | V |

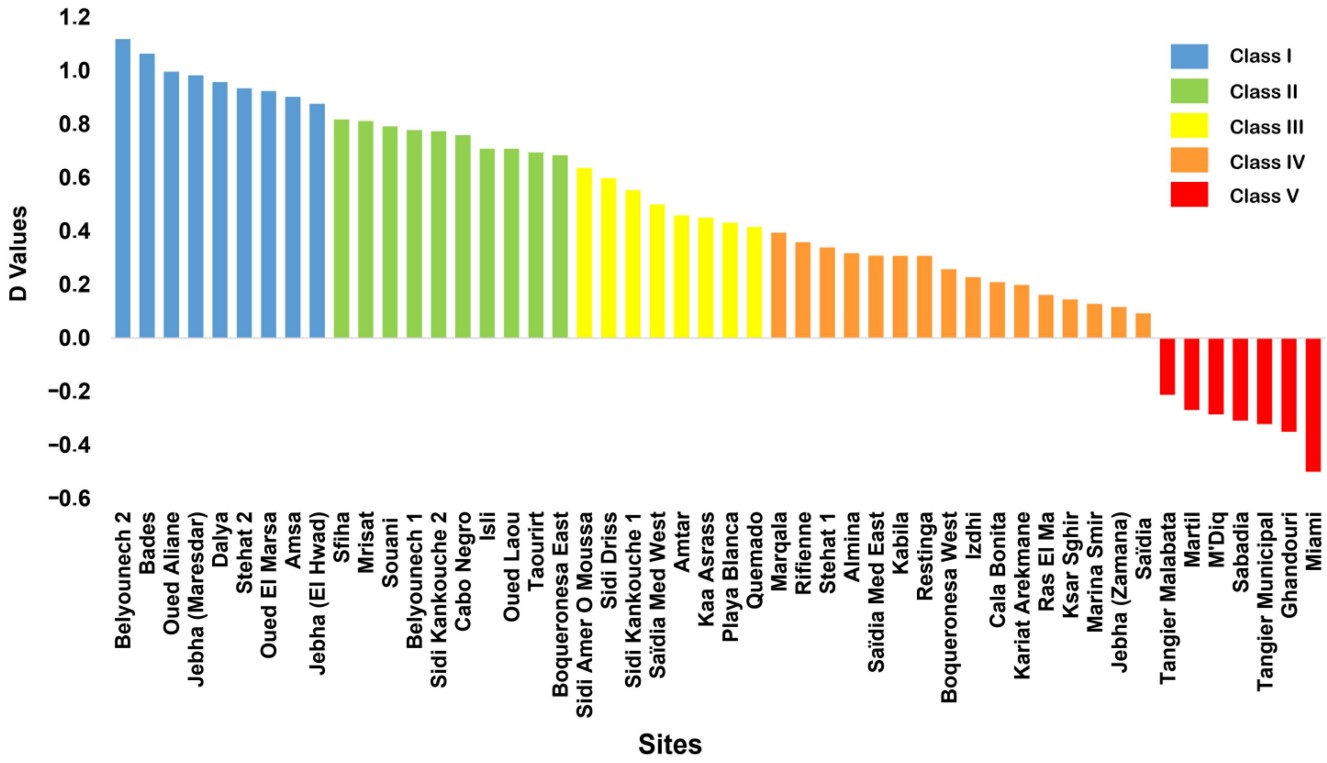

**Figure 3.** D scenic values for 50 sites along the Moroccan Mediterranean coast.

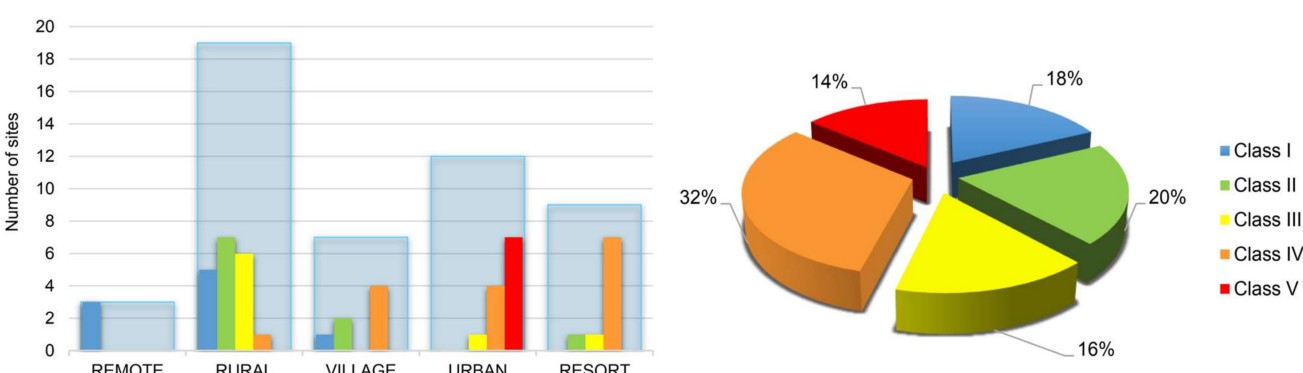

**Figure 4.** Coastal scenery percentage distribution by class and by type of beach along the study area. Blue rectangles indicate the total number of sites within each beach typology.

Practically, a curve inclined to the right reflects a high landscape quality due to the low rating on attributes 1 and 2 (Figure 5B, e.g., El Hwad); while, a curve inclined to the left reflects the opposite (Figure 5B, e.g., Tangier Municipal). High attribute values, 4 and 5, reflect the positive impact of the natural and human parameter (Figure 5C, e.g., El Hwad); while lower attribute values (1 and 2) reflect the negative impact of the natural or anthropogenic parameter (Figure 5C, e.g., Tangier Municipal) [20,27].

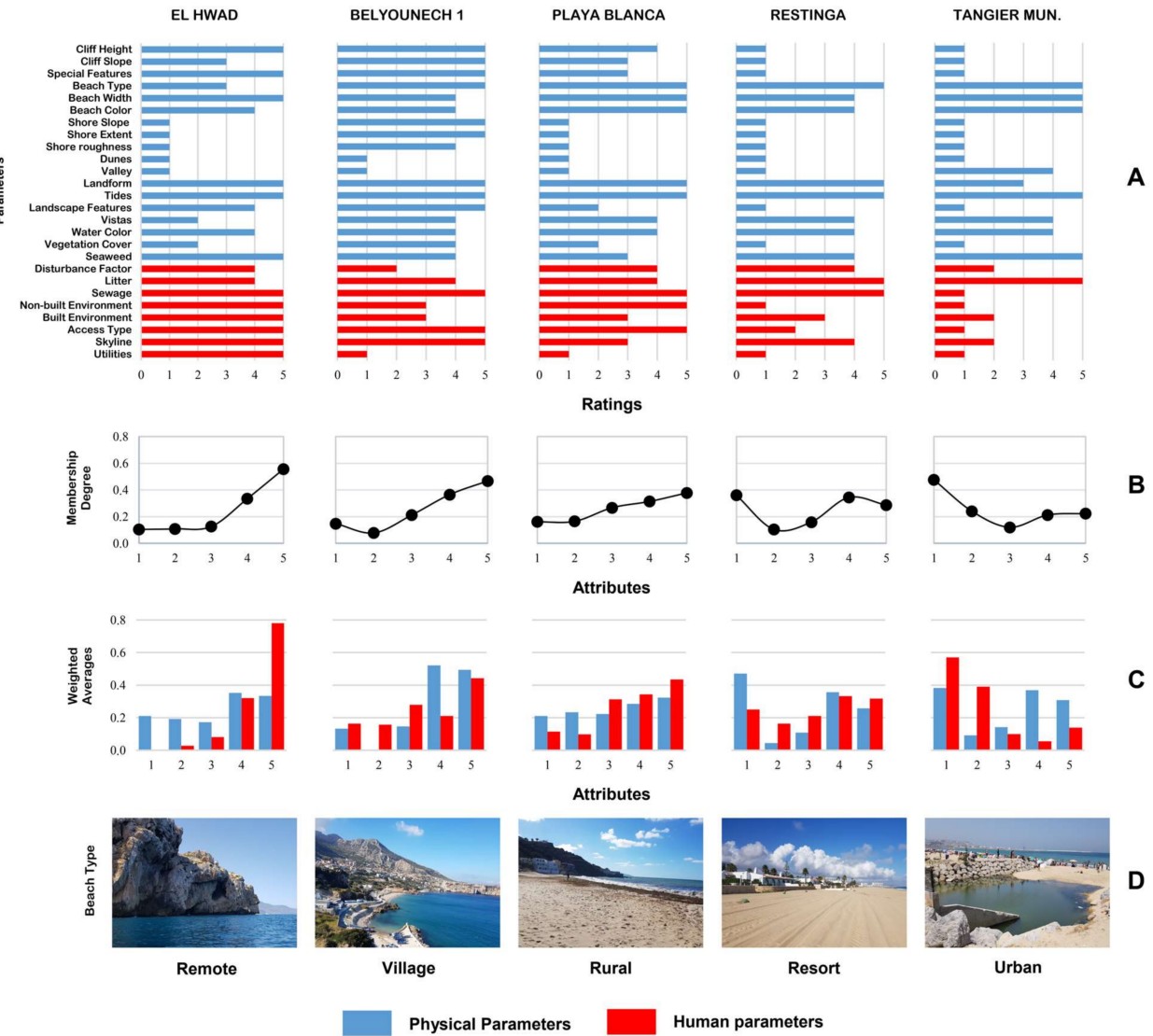

**Figure 5.** Examples of the site characteristics and status of the physical and human parameters for each class from I to V (left to right), with their respective (**A**) scenic evaluation rating histograms, (**B**) membership degree curve, (**C**) scenic histograms of the weighted averages, and (**D**) beach typology for El Hwad (Class I, Remote), Belyounech 1 (Class II, Village), Playa Blanca (Class III, Rural), Restinga (Class IV, Re-sort), and Tangier Municipal (Class V, Urban).

### 4.1. Classification and Characteristics of Sites

Analysis of the scenic evaluation value (D) obtained for each one of the 50 sites investigated along the Moroccan Mediterranean coast, as well as the sites characteristics and distribution, are presented in following lines:

#### 4.1.1. Class I

These are natural sites with a very high landscape value (D ≥ 0.85). Along the study sites, 9 out of 50 beaches belong to this class, or 18% of the study sites (Table 2 and Figure 4). Sites are distributed among rural (5), remote (3), and village (1) areas (Figure 4). Class I sites are located in protected natural areas, or remote areas—this is often a requirement to achieve Class I [22].

Beaches belonging to this class are highly ranked due to the exceptional characteristics of the natural and human parameters that provide excellent coastal landscapes. High scores

obtained are essentially the direct result of minimal human influence [50], which does not affect the natural aspect of the landscape. Further, their particular geological characteristics favour the formation of a multitude of attractive natural features, cliffs, coastal landscape features, valleys, and mountainous reliefs [27]. All these sites without exception have high-altitude cliffs with very remarkable special features and a close to vertical cliff slope (Parameters 1 to 3, Table 1), often accompanied by smoothed or flattened rocky shore platforms covering extensive areas (Parameters 7 to 9, Table 1). Other parameters, such as water clarity and colour, natural vegetation cover, noise disturbance, litter, sewage, and built and non-built environment, recorded scores of 4 or 5 (Parameters 14; 16; 17; 19 to 23, Table 1).

Belyounech 2 (B13, Figure 6A) was ranked first because of its natural richness and outstanding beach features. This site presents an attractive landscape with strong land–sea contrasts. The coastline is surrounded by mountains and formed by alternation of cliffs and small wild beaches. The beach is made up of small bleached gravel, and the water colour is very clear blue with a low presence of vegetation debris. The human parameters are highly rated, indicating very low human impacts.

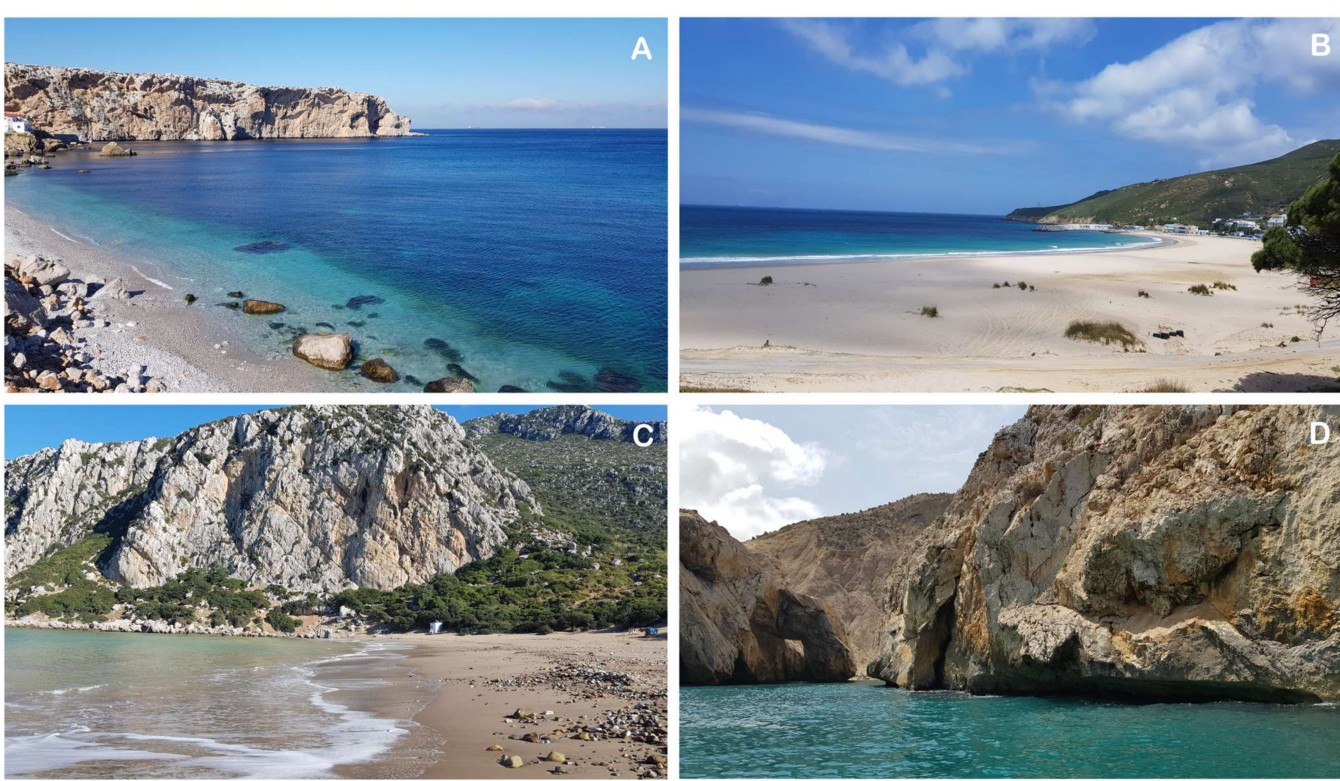

**Figure 6.** Examples of Class I sites, showing their exceptional natural and landscape characteristics ((**A**), Belyounech 2; (**B**), Dalya; (**C**), Oued El Marsa; (**D**), Maresdar).

The sites of Dalya (B11, Figure 6B) and Oued El Marsa (B12, Figure 6C) are two nearby sites located on a small bay between two high-cliff capes. They achieve high scores for cliff parameters, beach type and width, and rocky shore platform. Most human parameters obtained the maximum attribute values (5). Dalya beach retains its wild and natural appearance, presenting well-developed dune fields and dense vegetation. Clean-up activities are there, carried out within a national program, initiated by the Mohammed VI Foundation for the Environment since 2015, and has proudly held Blue Flag status for many years.

Maresdar (B30, Figure 6D) is a bay beach with excellent panoramic coastal views. It is a gravel beach, with rocks, caves, arches, and coves. The site is located in a natural landscape devoid of litter and sewage, with an absence of human occupation and anthropogenic

structures. The natural parameters are highly rated, and almost all human parameters have a score of five (excellent).

Examples of this class around the world are the beaches of Long Reef, Australia [46]; Zaglav, Croatia [19]; Çirali, Turkey [32]; Playa de los Genoveses, Spain [33]; Cabo de la Vela, Colombia [49]; Playa Sirena, Cuba [34]; and Ahu Tongariki, Chile [50].

### 4.1.2. Class II

These are attractive natural, semi-natural sites with high landscape values ($0.85 > D \geq 0.65$) and very acceptable anthroAlonpogenic impacts [33,37]. g the study area, 10 out of 50 beaches, or 20%, were classified in this category (Table 2 and Figure 4). They are distributed among rural (7), village (2), and resort (1) areas (Figure 4). Most sites are rural areas far from urban centres where human influence is low. These coastal areas are classified less than Class I because of a lower scoring of natural parameters [27,50], and/or good scores of physical parameters but intermediate scores at some human parameters such as noise and litter (during summer time), presence of beach facilities, and built and no-built environments (e.g., Belyounech 1, Figure 5). Class II sites have particular characteristics, such as the presence of cliffs or rocky shores (Parameters 1 to 3; 7 to 9, Table 1). They maintained high scores for beach characteristics such as typology (sand), width and colour, presence of dune fields, water colour and clarity, and natural vegetation cover (Parameters 4–6, 10, 16, and 17; Table 1). Unfortunately, the negative aspects included mainly noise disturbance, presence of few scattered litter items, and a certain degree of human occupation (Parameters 19, 20, 23, and 26; Table 1). As an example, the site of Sfiha (B38, Figure 7A) is a coast of high landscape value. The beach is endowed with very important tourist potential; a series of coves and islands are located a few meters from the beach and offers an attractive panoramic view. This site is not Class I due to the absence of a rocky shore, the dark colour of the beach sediment, and a reduced beach width. In addition, the site is currently experiencing the development of a series of tourist complexes, which directly influences Parameters 23–26 (Table 1).

The site of Cabo Negro is a seaside resort among a mountain landscape (B21, Figure 7B). The natural parameters are highly rated. The landscape is characterized by the presence of cliffs covered by dense vegetation. The negative aspects affect more the human parameters, with the development of tourist settlements on the cliff in strong disagreement with the natural landscape (Figure 7B). The site still preserves its natural aspect because tourist activities take place only in the summer period, when it receives a significant number of visitors. Beach users practice seaside activities from May to September, but August and July are the months when beaches receive the highest number of local, national, and international tourists [51].

Examples of beaches with similar D values in the world are Tojo Beach, Japan [46]; Haven Beach, USA, and Cliffs of Moher, Ireland [32]; Punta Camarones, Colombia [20]; Sandals, Cuba [34]; and Hanga Kioe, Chile [50].

### 4.1.3. Class III

These are sites with very few exceptional landscape elements ($0.65 > D \geq 0.40$), attractive, but with considerable anthropogenic impacts [33,37]. Along the study sites, 8 beaches belong to this category, or 16% of the sites studied (Table 2 and Figure 4). They are distributed between rural (6), urban (1), and resort (1) areas (Figure 4).

Practically, Class III sites are often the result of increased anthropogenic pressures and show intermediate scores for physical and human parameters [33,37]. The major problem for the majority of these sites is the presence of omnipresent litter, usually scattered items linked to beach users, mainly local and national visitors (Parameter 20, Table 1). It has also to be mentioned the presence of intensive urban developments, primarily manifested by the construction of illegal buildings, or the installation of random unattractive tents, or unattractive public structures (Parameters 23, 25, and 26; Table 1). Such impacts were also

observed by Rangel-Buitrago et al. [49,50] and Anfuso et al. [27,34] along the Caribbean Sea of Colombia and Cuba, and along the South Pacific Ocean in Chile.

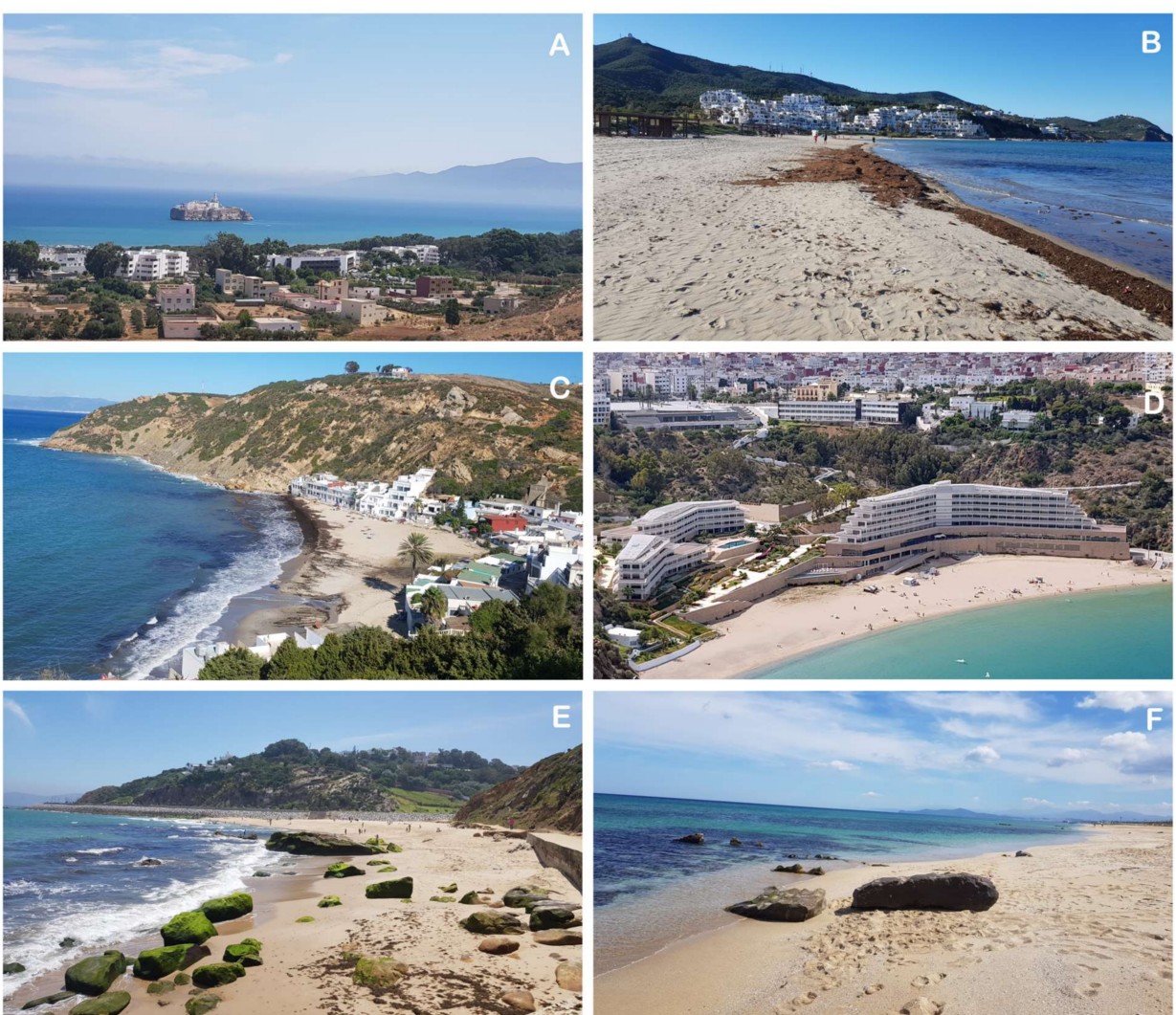

**Figure 7.** Examples of the characteristics of the physical and human parameters of Class II sites ((**A**), Sfiha; (**B**), Cabo Negro), Class III ((**C**), Playa Blanca; (**D**), Quemado), and Class IV ((**E**), Marqala; (**F**), Rifienne).

Within this study, the best examples of Class III sites are Playa Blanca (B6) and Quemado (B35). Playa Blanca (Figures 5 and 7C) is a small natural coastal sector located in a beautiful bay. This site is named Playa Blanca (i.e., white beach) because of the white colour of the sand. The beach is surrounded by spectacular cliffs, which are occupied by several unattractive buildings not in accordance with the natural landscape. Quemado is the most popular bay in Al Hoceima (Figure 7D). It is a seaside site located in an urban area in the middle of a cliffed coastline with a rich natural vegetation cover. The physical parameters obtained high scores, but this was not the case for the human parameters. Quemado was the highest rated urban beach.

4.1.4. Class IV

These are practically unattractive urban sites, with a low landscape value ($0.40 > D \geq 0$). A total of 16 out of 50 coastal sites achieved this category, or 32% of the study sites (Table 2 and Figure 4). Class IV is composed of the largest number of sites, distributed mainly among

resort (7), urban (4), village (4), and rural (1) areas (Figure 4). The scenic value of these sites is heavily lowered by considerable anthropogenic activities and impacts [34,46,50].

Two groups of beaches can be distinguished in this category: seaside resorts, and urban and semi-urban sites.

(i) The first group consists mainly of resorts that have unattractive coastlines. These sites generally have low scores for physical parameters and mediocre human parameters. The low values were due to low scores for cliffs, rocky shores, dunes, and natural vegetation. With the exception of the built/non-built environment, and public facilities (Parameters 22, 23, and 26; Table 1), the human parameters are highly rated, as sites are located at some distance from urban centres. No litter or sewage were observed, and beaches presented regular cleaning and maintenance activities.

Rifienne (B15), Almina (B16), Restinga (B17), Kabila (B18), Marina Smir (B19), etc., are excellent seaside resorts along the Tetouan littoral (Figures 5 and 7F). This coastline is characterized by a diversity of natural landscapes and outstanding physical characteristics. It is considered one of the best seaside resorts on the Moroccan Mediterranean coast and represents a very privileged destination that receives a large number of visitors. The above areas are characterized by the massive presence of seaside resorts, such as those observed in Varadero (Cuba) for Class IV [27].

(ii) Sites in the second group are urban and semi-urban areas and are not very attractive due to anthropogenic pressures. They have been well rated for their natural aspects, such as the presence of cliffs, the colour and clarity of water, and their natural vegetation. Negative aspects are mainly related to human parameters dominated by increasing urbanization, waste, sewage in some cases, noise, and utilities (Parameters 19–23 and 26; Table 1). These sites are in the migration process to become Class V sites. Examples are Marqala (B1) and Calabonita (B36).

Marqala and Calabonita are two beaches close to the city centres of Tangier and Al Hoceima. The site of Marqala is a small coast of several coves (Figure 7E); the beach is triangle-shaped, with fine clean golden sand and clear blue water, bordered by high cliffs covered by dense vegetation with an extensive flattened rocky platform. The site of Calabonita is a small cove, surrounded between two rocky limestone cliffs. Both sites are similar; their location close to urban centres greatly attenuates the human parameters, such as the presence of sewage.

4.1.5. Class V

These are unattractive urban sites with intensive development associated with very high human occupation [27,42], and low landscape values (D ≤ 0). In fact, these urban sites are considered unattractive according to the CSES classification, since this methodology is devoted to natural areas and evaluates the natural landscape and the degree of human intervention; therefore, many urbanized areas obtain low scores. The important characteristic is that the buildings, etc., blend into the surroundings and do not stand out, e.g., Cinque Terre (Italy) and Balneario Camboriu and Copa Cabana (Brazil). So, such areas can be much appreciated places, regardless of showing tall buildings or intense urbanization. The same principle applies here in this study, as sites such as Tangier Municipal, M'Diq, and Martil are very popular but their natural classification is scoring low.

This category includes seven beaches or 14% of the study sites (Table 2 and Figure 4), all of them corresponding to urban beaches (Figure 4). Natural and human parameters present low to very low values. The main responsible factors are human pressures, such as unattractive urbanization, a large amount of litter, sewage, poor horizon quality, absence of non-built environment, and noise disturbance (Parameters 19–26; Table 1).

On the northwest of Morocco, the bay of Tangier includes three urban beaches: Municipal (B2), Malabata (B3), and Ghandouri (B4) (Figure 5, Figure 8B,C and Figure 9A,C). The particular geographical position of the Bay between the Mediterranean Sea and the Atlantic Ocean makes this coastal landscape of huge potential interest. The negative aspects included urbanization—i.e., high buildings visible from great distances—evidence of

sewage at the Tangier Municipal site, and considerable quantities of litter at the Tangier Malabata and Ghandouri sites. Low scores are also due to the existence of human activities and structures (Figure 8).

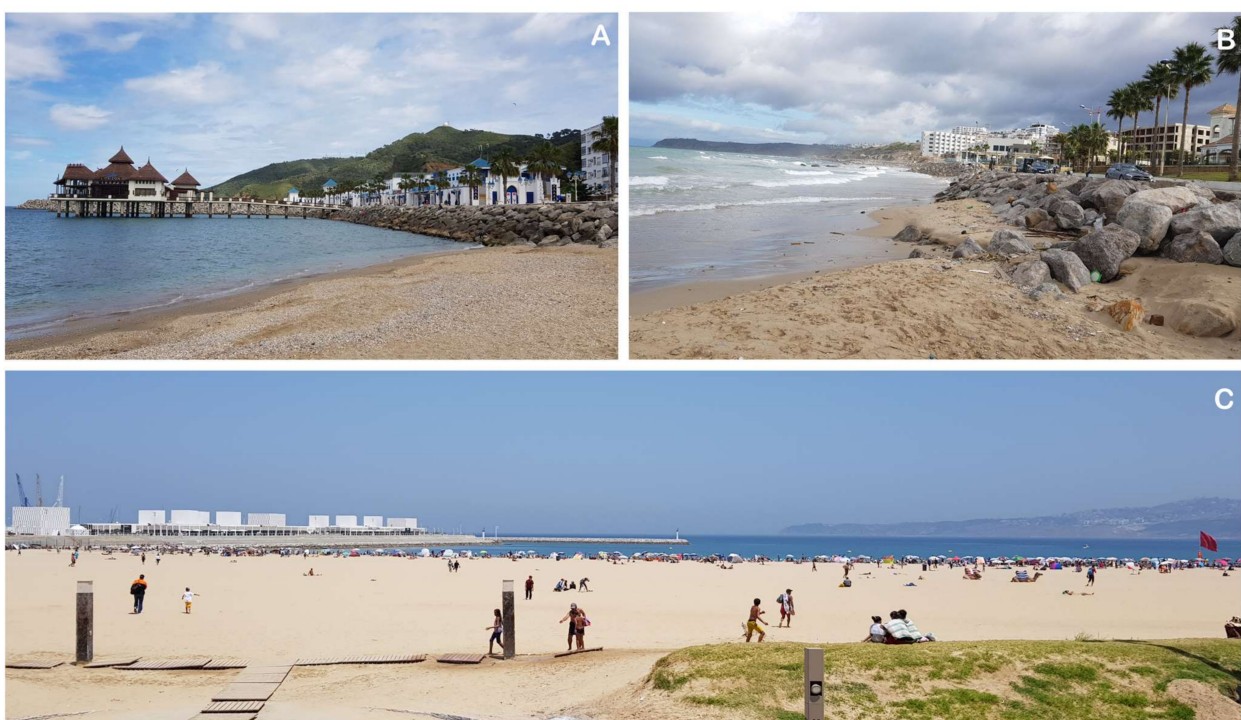

**Figure 8.** Examples of Class V sites showing some landscape features of urban areas such as buildings and the presence of public facilities ((**A**), M'Diq; (**B**), Ghandouri; (**C**), Tangier Municipal).

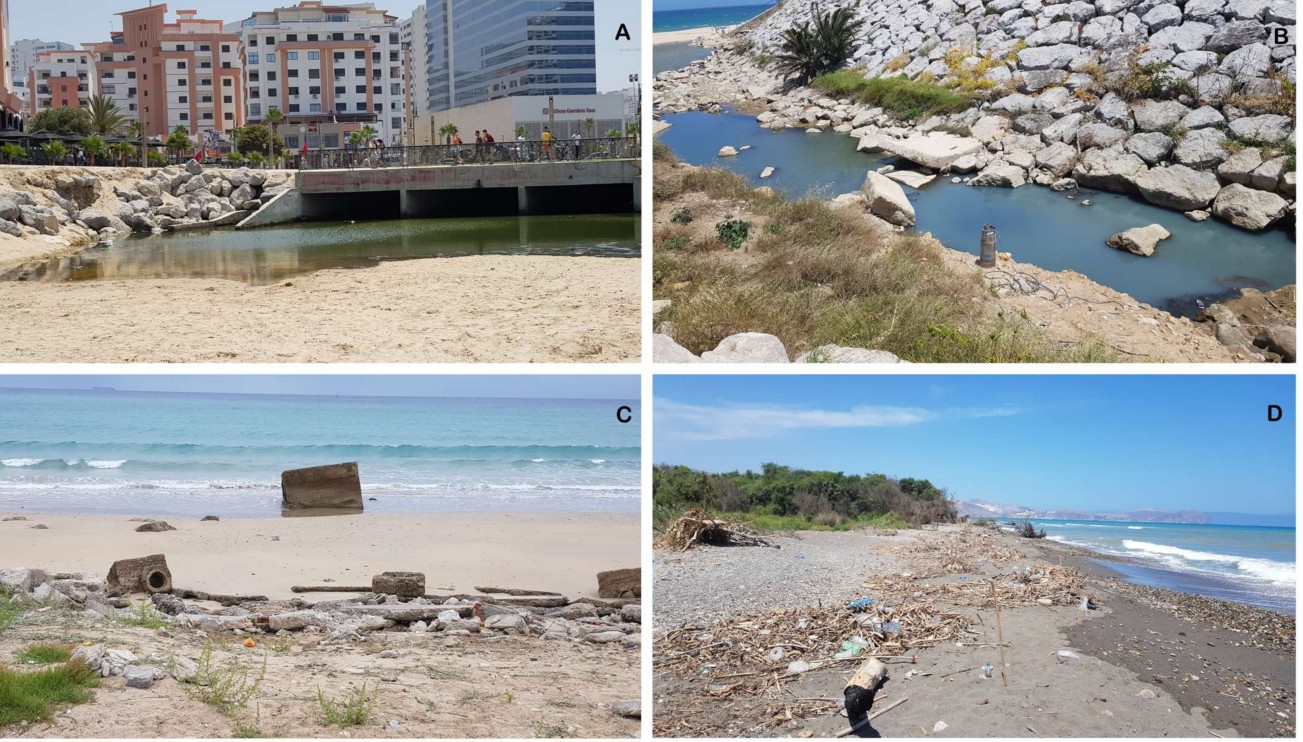

**Figure 9.** Examples of litter and sewage negatively impacting the beauty of coastal scenery ((**A**), Tangier Municipal; (**B**), Marqala; (**C**), Tangier Malabata; (**D**), Souani).

Coastal protection structures often promote, in downdrift areas, coastal erosion, which has a negative effect on the coastal landscape by reducing beach width and favouring large size sediments [52].

The sites of Mdiq (B20) and Martil (B22) are two urban areas belonging to the Tetouan coastline. Natural parameters at Mdiq and Martil sites are poor and mediocre, respectively. Both sites obtained good scores concerning beach characteristics such as nature and colour of sediments, beach width, plus a mountainous skyline landform for M'Diq (Figure 8A), and the presence of well-developed dunes, natural vegetation cover, and a river valley for Martil. Human parameters are very low, considerably reducing the quality of the landscape.

*4.2. Considerations for Moroccan Coastal Management*

Along the Moroccan Mediterranean coast, 38% of the sites studied are in Classes I and II, 18% in Class III, while the lower classes (IV and V) presented 46% of the sites studied (Table 2, Figure 4). The considerable percentage of Class IV and V sites, corresponding to almost half of the total amount, clearly reflects the degradation that the coastal landscape along with the Moroccan coast has suffered, as a result of an accelerated littoralization, and the lack of adequate management that led to the destruction of coastal environments along a large part of the area investigated.

This situation highlights the limits of the laws and regulations that govern the marine environment in general and the coastline in particular, as mentioned by Nachite and Sbaï [41]. In this regard, Morocco has established a set of legal instruments to adequately manage its coastline. The most important are (i) Law No. 81-12 of 2015, relating to the coastline, aiming, among other things, to regulate construction, to prevent and reduce pollution, and prevent degradation of the coastline; (ii) Law No. 28-00 of 2006, relating to waste management and disposal, which aims to limit the negative impacts of waste on human health and the environment; and (iii) Law No. 36-15 of 2016, concerning water quality, establishing—as a tool to reduce water pollution—the legal framework for the establishment of sewerage networks and wastewater treatment plants.

In addition, Morocco has signed several regional and international conventions on the protection of the marine environment. Among others, the Barcelona Convention and especially protocols as the "Integrated Coastal Zone Management" (Ratified by Morocco on 21 September 2012) and "Telluric" ("Land-based Sources and Activities of Pollution") (Approved by Morocco on 26 November 1996), the London Conventions 1972 (Ratified by Morocco on 18 February 1977, Protocol 1996 ratified on 25 February 2016), MARPOL 73/78—as amended by the 1978 Protocol (Approved by Morocco on 12 October 1993)—and Basel conventions (Approved by Morocco on 28 December 1995). Furthermore, Morocco has launched programs and initiatives to properly manage and protect its coastline. The most important are (i) the National Coastal Plan (PNL) (2020), which aims to reconcile environmental protection and economic activities; (ii) the National Household Waste Program (PND) (2008), which provides for the collection and cleaning of household waste; and (iii) the National Program of Sanitation and Wastewater Treatment (PNA) (2005), which is one of the most relevant instruments implemented to safeguard the coastline. In terms of monitoring, Morocco, within the framework of the "Telluric" protocol, ensures the implementation of the MEDPOL program (UNEP/MAP) (2013), relating to the assessment and control of marine pollution on the scale of Morocco; in addition, since 2001, the Mohammed VI Foundation for Environmental Protection (M6FEP) has launched the "Clean Beaches" program.

In Morocco, the national ICZM strategy was launched in February 2005, as the main axis of the draft law relating to the Littoral (a Law Project mentioned above). The approach was reinforced in January 2008 with the signing, alongside thirteen other countries, of the 7th protocol of the Barcelona Convention on ICZM and the ratification of this protocol in 2012. Integrated Coastal Zone Management in Morocco is considered the major tool for the implementation of sustainable development in coastal areas. Today, it constitutes a framework and a reference for the management of many coastal areas. Indeed, recently,

on 5 May 2022, the Moroccan government adopted draft decree no.2.21.965 relating to the National Coastal Plan, which supplements the implementing texts of Law 81.12 related to coast management. The main aim of this plan is to integrate the dimension of coastal protection into sectoral policies, particularly in the areas of tourism, housing, industry, and infrastructure.

In addition, this type of study is closely compatible with Integrated Coastal Zone Management (ICZM) strategies, as it provides a wealth of data that allow coastal resources to be managed in a sustainable and reasonable manner in order to answer questions related to the requirement for effective coastal management. In fact, the combination of coastal scenery management and ICZM strategies is an excellent opportunity to strengthen several integration efforts. Adequate management and conservation of coastal areas require that the current use of coastal resources meet the needs of the local populations. Any coastal scenery management strategy requires the implementation of effective and efficient solutions based on scientific knowledge and includes the priorities and preferences of coastal users [53]. Results obtained in this paper respond adequately to previous questions.

Therefore, results on the Moroccan coastal landscape assessment represent a powerful tool that contributes to the sustainable development of coastal areas under the aegis of ICZM/MSP. They allow to identify the main problems that affect the coast, to contribute to the implementation of strategies from a local, regional, and national perspective, and also provide invaluable background information and a sound scientific basis for any coastal development plan envisaged. In addition, several countries have successfully exploited this type of information to build their own ICZM strategies (e.g., Spain, Cuba, Chile, etc.).

Results obtained within this paper show that, despite this legal arsenal, it seems that environmental and social aspects are often poorly or not taken into account during the design and implementation of coastal space management systems. In this respect, the results presented, and the proposed solutions of improvement, are of great help to coastal managers and stakeholders to overcome degradation problems along the Moroccan Mediterranean coast. Since this methodology is universal and can be applied to many coastal areas around the world, the results of this study are also very important for the scientific community, especially for coastal areas that are similar in terms of the characteristics of physical and human parameters and nature of the human interventions.

As an example, it is possible to state that:

- Class I: All sites are natural and very well rated, but still retain their wild aspects. These sites require conservation and precautionary measures to avoid any future development of constructions and infrastructures, and the adaptation of strong adequate management measures to avid the presence of solid waste and sewage.
- Class II: Some sites can be improved in terms of certain parameters to be upgraded to Class I. In this respect, solid waste appears to be the most degrading factor; a regular program of beach cleaning and maintenance (daily or weekly) could improve the ranking of these beaches. If beach litter is reduced in Mrisat (D = 0.81), Souani (D = 0.79, Figure 9D), and Taourirt (D = 0.69), D values, respectively, increase to 0.95, 1.07, and 0.87, and thus all sites upgrade to Class I. The same applies to Sfiha beach (D = 0.82), which could be improved and move to Class I by limiting the impact of litter, noise, and tourist infrastructures.
- Class III: Here again, it appears that the problem of solid waste is the main factor of degradation. These sites (Sidi Amer O Moussa, D = 0.64, and Sidi Driss, D = 0.60) represent rural areas, with great landscape values and are mainly devoted to agriculture activities and artisanal fishing; if regular cleaning activities are carried out they could be easily upgraded to Class II.
- Class IV: Sites in this class are very close to large urban centres and directly affected by their impacts, namely, a high number of visitors, especially during the summer period, which results in a high level of noise disturbance. Other factors are the presence of sewage, as observed for Marqala (Figure 9B) and Calabonita.

- Class V: This class presents beaches located in highly urbanized areas. Litter, sewage, noise disturbance, and facilities are the main factors of degradation (Figure 9A,C). It is difficult to improve the impact of infrastructure, especially in terms of cost, but other factors can be easily improved.

The presence of beach litter is relevant at Martil, Miami, Sabadia, Tangier Malabata, Ghandouri, Kaa Asrass, and Stehat 1, which were rated 1 or 3 in Parameter 20 (Litter, Table 1). Wastewater assessment results were good with the exception of Tangier Municipal, Martil, Mdiq, Sabadia, Calabonita, Marqala, and Miami, where values where scores from 1 to 3 were recorded (Parameter 21, Table 1). Other sites are highly rated due to appropriate wastewater management actions.

In addition, this type of assessment makes it possible to monitor the landscape quality of beaches and to measure the degree of effectiveness of the management measures. In this context, six sites of this study have been assessed previously, and by the same methodology, by Khattabi et al. [38] and Williams and Khattabi [39], namely, Taourirt, Boqueronesa, Kariat Arekmane, Sidi Driss, Ras El Ma, and Miami. The comparison—after over a decade—shows that the Miami site is the only one experiencing degradation, moving from Class IV to Class V (D = 0.09 in 2009 to D = −0.50 in this study). Degradation was linked to over-occupation and over-use because an expansion of industrial and port constructions. The other sites maintain their rankings, with a slight decrease in D values for the Taourirt and Sidi Driss sites and a partial increase for the Kariat Arekmane and Ras El Ma sites.

It should not be forgotten that surveys of beach users remain essential since adequate management of coastal sites essentially involves an understanding and awareness of users' priorities and expectations (Figure 10). Starting from those priorities, assessment of natural and human parameters allows for the selection and characterization of the essential elements that must be taken into account in a sound management plan.

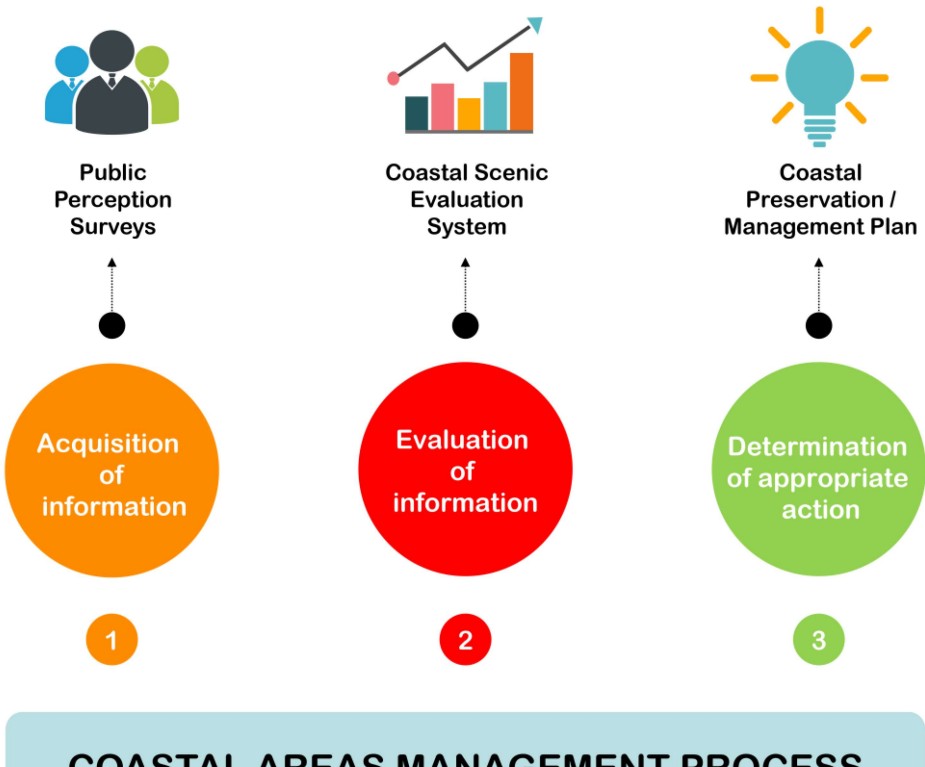

**Figure 10.** Determination of priorities and appropriate actions through the coastal zone management process.

Practically, litter and sewage are among the main degradation factors of the Moroccan Mediterranean coast (Figure 9), as exemplified in Rangel-Buitrago et al.'s [54] analysis of the effects of litter on scenery in Colombia; this represents a global environmental problem that affects all coastal areas [50,55]. During a study of Nador province (on the Mediterranean coast of Morocco), sewage and litter were found to be the main obstacle for beaches to upgrade to a higher class, and many excellent beaches were strongly influenced by them [39]. Solid waste has been recognized for about 60 years as a serious environmental pollution problem, and its impact is of great global importance [56–58]. In addition, studies have defined the Mediterranean coasts as one of the areas most affected by marine litter on a global scale [59,60]. For this, the management priority on the Moroccan coastline must be for litter, which constitutes a source of visual pollution with real impacts on the quality of the landscape [20,50], and on the local and national economy [61–64]. The same principle applies to wastewater management, which requires a great deal of attention in terms of proactive actions by taking the necessary measures.

The persistence of litter and sewage as factors of degradation, linked to the intensive littoralization and anarchic occupation of the coastline, is mainly observed in peri-urban areas, where litter collection and the sewer network are absent or failing [41,65]. The present situation should call on the Moroccan authorities for an urgent review of the "National Household Waste Program" (2008) and the "National Program of Sanitation and Wastewater Treatment" (2005). These programs should include beach solid waste management, taking into account the sources of waste and regular cleaning throughout the year [66,67]; and the eradication of untreated wastewater discharges.

In addition, and knowing that municipalities are responsible for municipal waste and sewage management at institutional and organizational levels, these programs must incorporate appropriate training plans, especially aimed at local managers and technicians responsible for waste collection programs and sanitation network management [67,68]. The results of this study also aim to adopt an appropriate management of coastal areas, to conserve coastal resources and, consequently, the attractiveness and tourist potential of the Moroccan Mediterranean coast, in accordance with the strategies and programs launched by the Moroccan authorities on the development and sustainability of the seaside tourism sector. It should be noted here that one the recommendations of the ESEC (Economic, Social and Environmental Council) "For a new national tourism strategy" (B.O. 7080 of 7 April 2022) is the development of sustainable and socially responsible standards. This study also represents an opportunity for beach users around the world to know more about the diversity of landscapes observed along with the Moroccan Mediterranean coast and their characteristics and value. This is very relevant considering that Morocco is located in a region of the world with a high level of tourism development, i.e., the Mediterranean. Consequently, this type of study aspires to accompany this development by providing enough information to the actors and managers to improve the quality of the coastal areas and their associated tourist offer.

At the same time, communication and awareness plans for beach users, and particularly managers of sports and leisure services and activities, need to be strengthened. This is to prevent and limit the generation of beach litter related to users [67] and the noise impacts related to recreational and sports services and activities. It should be noted here that, concerning noise pollution, even if the law No. 11-03 of 2003 on the protection and development of the environment recognizes this kind of pollution in its article 47, no regulations, or standards, have been defined so far, apart from noise pollution in the workplace. Finally, the question of improving the physical parameters remains very complicated, and practically impossible to modify or change the natural characteristics [20,39], especially in urban areas, as they are difficult to manage and their transformation is very costly and linked to external factors, which further increases the difficulty. The most appropriate approach in this case is to focus on ways to improve human parameters.

## 5. Conclusions

The landscape is a fundamental element of coastal tourism and an important driver of local and national economic development. Landscape assessment is an important tool to enhance the potential for tourism development and improve the quality of beaches and the maintenance and conservation of the coast. Along the Moroccan Mediterranean coastline, fifty sites were studied for the assessment of the quality of beaches using a coastal scenic evaluation system.

Class I and II sites are highly rated for the majority of natural and human parameters and have excellent coastal landscapes with outstanding features, and low influence of human pressures. High values were recorded in highly attractive natural, protected, and remote areas. The main parameters evaluated were the presence of cliffs, rocky shores, landscape features, water clarity and colour, vegetation cover, absence of noise disturbance, absence of waste and wastewater, as well as the quality of the built and non-built environment.

Class III, IV, and V sites are progressively lower rated, represented by village, semi-urban, and urban areas, with landscape values heavily affected by considerable anthropogenic activities. The lowest scores were recorded in very urbanized areas of large cities. Such coastal areas obtained negative D values, representing extremely degraded unattractive landscapes. The negative aspects affect more the human parameters, such as unattractive intensive urbanization, noise, large amount of waste, sewage, absence of the non-built environment, and poor quality of the skyline.

Results obtained in this paper show a certain degree of degradation of the coastal landscape that the Moroccan coastline has suffered because of the human pressures and, therefore, the need for adequate management actions. It is rare that managers can intervene with the physical parameters, but they can with the human ones. In this case, management should try to protect the natural aspects of the coastal landscape by laws, in the natural places where these parameters are still in a favourable state of conservation. This means that management attentions and efforts should be focused on improving the human parameters to mitigate the impact on coastal landscapes.

Morocco has made efforts to protect its coastline through numerous international conventions. Currently, it has a set of legal, especially Law No. 81-12 of 2015 and Law No. 28-00 of 2006, and institutional tools that are theoretically largely adequate and appropriate to manage coastal areas properly. It is clear that further efforts need to be made through strategies, programs, and management plans by local administrations to rehabilitate degraded coastal areas of the Moroccan coast. For this reason, management and protection of coastal areas on a Moroccan scale has become a fundamental question in the framework of effective environmental governance that puts in place a sustainable development perspective.

**Author Contributions:** Conceptualization, D.N., N.E.-R.; methodology, D.N., N.E.-R.; software, N.E.-R.; validation, D.N., G.A., A.T.W.; investigation, N.E.-R.; data curation, N.E.-R.; writing—original draft preparation, N.E.-R., D.N., G.A., A.T.W.; writing—review and editing, N.E.-R., D.N., G.A., A.T.W.; supervision, D.N., G.A.; project administration, D.N. All authors have read and agreed to the published version of the manuscript.

**Funding:** This research received no external funding.

**Data Availability Statement:** Data supporting reported results can be found asking directly of the second author.

**Acknowledgments:** This work is a contribution to the PAI-Research Group RNM-328 of Andalucía, Spain.

**Conflicts of Interest:** The authors declare no conflict of interest.

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
