# Peer review of "Coastal Scenic Quality Assessment of Moroccan Mediterranean Beaches: A Tool for Proper Management"

_water, doi:10.3390/w14121837_

Round 1

Reviewer 1 Report

Dear authors,

I found the manuscript interesting and with sound results. I will be very pleased to see it published in its final form in the journal Water.

I have only one remark and it is related with the way figures and tables are presented.

The legends (captions) of figures and tables should be revised. I suggest a detailed revision of all the captions, these must be the most complete as possible, because figures should be legible without the need to see the main text. For instance, the tables and figures captions do not refer the study area where or the study context.

I would like to see this considered in a final version of the manuscript.

Regards

Author Response

Thank you for your comments, please see attached file.

Reviewer 2 Report

The manuscript is interesting and topical but it needs more work to make it scientifically stronger

The Introduction is generally good, but it should have clear links to the important processes of coastal zone planning and marine spatial planning. These themes are widely covered by recent scientific literature. Also 4.2. lacks citations to this literature, yet ICZM is mentioned in the legislative analysis.

The legends of the figures and tables are in most cases too short and may lack information that would help their interpretation.

The enlarged map windows in Figure 1 are not marked on the main map and are at different scales. The enlarged maps should also presernt some new information (e.g. land cover, population, hotels) to make their use justified.

The methods descriptions are too general. Steps 1 and 2 are sketchy, even misleading. Apparently all parameters are taken straight from sources 40 and 41. In such case, they provide methodological basis for this study but do not form parts of its own empiricism. The passive formulations should be replaced by clearer expressions. Also the remaining descriptions mainly reflect the broad methodology rather than tell how this study was conducted. The reader would expect more detailed information about the details of this Moroccan case study; for example, how the data were collected and classified at the practical level. You could even consider to prepare a series of photographs with examples of different landscapes and their classifications. The methodological descriptions should be detailed enough to allow someone to carry out a follow-up study on the same sites years later.

The BARE typology and how it is applied should also be better described. It is now mentioned just in very general terms.

Table 1 describing the coastal landscape assessment system is the same as Table 1 in [41], although some details are expressed slightly differently. The source of the information should be mentioned.

The five coastal landscape classes defined by their D-value intervals contain different ranges of numeric values. What are the boundaries based on?

Figure 3 is clear. However, the distributions of classes I to V could also be shown on a map. This would allow to take into account regional features and to discuss their implications in tourism and its development.

The sentence "Classification of ..." (between 4.1. and 4.1.1.1.1.) does not fit in this part of the manuscript.

Table 2 contains little relevant new information (Figures 1 and 3 present the same data) and could be omitted.

Figure 5: although simple in principle, the details in this figure get difficult to follow as the methodology is not adequately described. There are new concepts in the figure: attributes, membership degrees, weighted averages.

It is somewhat problematic to consider urban sites automatically as “unattractive” (4.1.5.). The fact that people are different and can have different likings deserves more attention. There are plenty of scientific literature about these types of questions.

The last part of the manuscript discusses selected legislative and management priorities. The writing style may be a bit lengthy and redundant. Authors could better take advantage of the opportunity to move away from the level of detail and return to the teasers at the beginning of the introduction, which discuss, among other things, the touristic values and development potential of the Mediterranean and Moroccan coasts upon their scenic attraction.

The conclusions should rather be shorter and more precise. Now they contain many details mentioned earlier and a low number of views of a wider scale.

Author Response

(The authors gave the same response as above.)

Reviewer 3 Report

This article synthetically describes the coastal landscape of northwest Morocco (essentially the Moroccan Mediterranean coast), with the aim of analyzing the different sources of impact and presenting useful complementary information for a possible future management plan. To this end, it applies a methodology known as the "Coastal Scenic Assessment System" (CSES).

The literature on the topic is abundant and this manuscript follows many other publications (most of the authors themselves) with identical methodology.

In fact, the methodology followed is commonly used in works with identical objectives, as demonstrated in several case studies published in the last two decades. Therefore, scientifically, this manuscript presents neither a new method nor a new methodology.

The results of this case study may be of interest and possibly useful to users and managers of a sizable ecological system, such as the Moroccan Mediterranean coast, but are of little value to the international scientific community.

Anyway, there are aspects to be improved, including (1) a more informative location map of the investigated sites (Figure 1); (2) basic details of the mathematical formalism of the CSES methodology, and (3) essential elements for evaluating the quality and credibility of the results and their possible reproducibility. Such elements must be provided, eventually, in an appendix or as supplementary material.

Furthermore, for a case study, 72 references are not justified, and 50% (36/72) of self-citations are unacceptable.

Author Response

(The authors gave the same response as above.)

Reviewer 4 Report

Dear Colleagues, thanks for this manuscript on coastal scenic quality assessment of Maroccan beaches.

I acknowledge that so much work was done with the round table discussion, but the manuscript presents scientific weaknesses.

Please put in introduction part: what is the purpose of this study, how does it enrich our knowledge on management measures?

Literature review: this part is missing.

Research methods are poorly described and are not linked to theory or conceptual framework since these are also missing.

Data analysis, consequently, is poor. Table 1 is difficult to decode.

Results and Discussion are poorly written and do not provide a strong statement or a critical analysis of the findings. Explain how we can use your results for other countries.

Author Response

(The authors gave the same response as above.)

Round 2

Reviewer 3 Report

The manuscript was punctually reworked; improved overall, but retains the most important shortcomings identified in a previous review. It falls within the scope of another case study, following many other publications (many of them by the authors themselves) that use identical concepts and the same methodology.

Although with a slight reduction in references and self-citations, 29 self-citations out of 69 references (25 by author Williams A.T.) remain unacceptable; a scientific article cannot be confused with a repository of self-citations.

Therefore, this manuscript hardly falls within the scope of a Scientific Article; at most, it may be acceptable as a Case Study.